# Scalable Functionalization of Polyaniline-Grafted rGO Field-Effect Transistors for a Highly Sensitive Enzymatic Acetylcholine Biosensor

**DOI:** 10.3390/bios12050279

**Published:** 2022-04-27

**Authors:** Dongsung Park, Dongtak Lee, Hye Jin Kim, Dae Sung Yoon, Kyo Seon Hwang

**Affiliations:** 1School of Biomedical Engineering, Korea University, Seoul 02841, Korea; dpark7047@gmail.com (D.P.); ehdxor11@korea.ac.kr (D.L.); 2Department of Clinical Pharmacology and Therapeutics, College of Medicine, Kyung Hee University, Seoul 02447, Korea; 3Institute of Chemical Process (ICP), Seoul National University (SNU), Seoul 08826, Korea; hyejinkim.mail@gmail.com

**Keywords:** reduced graphene oxide, field-effect transistor (FET), polyaniline, pH-sensing, acetylcholine, acetylcholinesterase, drug screening

## Abstract

For decades, acetylcholine (Ach) has been considered a critical biomarker for several degenerative brain diseases, including Alzheimer’s, Parkinson’s disease, Huntington’s disease, and schizophrenia. Here, we propose a wafer-scale fabrication of polyaniline (PAni)-grafted graphene-based field-effect transistors (PGFET) and their biosensing applications for highly sensitive and reliable real-time monitoring of Ach in flow configuration. The grafted PAni provides suitable electrostatic binding sites for enzyme immobilization and enhances the pH sensitivity (2.68%/pH), compared to that of bare graphene-FET (1.81%/pH) for a pH range of 3–9 without any pH-hysteresis. We further evaluated the PGFET’s sensing performance for Ach detection with a limit of detection at the nanomolar level and significantly improved sensitivity (~103%) in the concentration range of 108 nM to 2 mM. Moreover, the PGFET exhibits excellent selectivity against various interferences, including glucose, ascorbic acid, and neurotransmitters dopamine and serotonin. Finally, we investigated the effects of an inhibitor (rivastigmine) on the AchE activity of the PGFET. From the results, we demonstrated that the PGFET has great potential as a real-time drug-screening platform by monitoring the inhibitory effects on enzymatic activity.

## 1. Introduction

The chief neurotransmitter, acetylcholine (Ach), plays a vital role in both the central nervous and peripheral nervous systems [1]. Ach regulates a wide range of biological processes, including neuronal proliferation, differentiation, and apoptosis [2]. Recently, several studies revealed that imbalances in Ach levels in the nervous system are associated with several degenerative brain diseases such as Alzheimer’s, Parkinson’s disease, Huntington’s disease, and schizophrenia [3,4,5]. As aberrant Ach levels in the nervous system are deeply associated with these severe disorders, many researchers have sought to develop highly sensitive and accurate analytical biosensors for monitoring local Ach concentrations, although conventional techniques for Ach detection have been explored, including liquid chromatography [6], electrochemiluminescence [7], and screen-printed electrochemical biosensors [8]. The capacity of these methods for scalable fabrication and sensitive Ach detection is still limited by their time-consuming procedure, the requirement for labeling agents, lack of sensitivity or dynamic range, and high costs.

Among the diverse bioelectronic transducers, nanomaterial-based field-effect transistors (FETs) have attracted considerable attention for developing label-free and highly sensitive biosensors. In particular, graphene has been considered an ideal channel material for constructing ultrasensitive FET biosensors due to its intrinsic properties such as outstanding electrical conductivity, chemical stability, high carrier mobility, and flexibility [9,10,11,12]. Numerous studies have been exploiting graphene-based field-effect transistor (gFET) biosensors for monitoring pH variation, cellular metabolisms, hybridization of deoxyribonucleic acid (DNA), antigen-antibody bindings, and enzymatic reactions [13,14,15,16]. The detection mechanism of gFETs is based on changes in the electrical conductance or the charge neutrality point, called Dirac point, generated by the doping effect in the graphene channel. Several groups have recently reported that enzyme-immobilized gFET biosensors can detect several analytes such as glucose, penicillin, and Ach by an enzyme-substrate reaction [17]. In detail, the reaction induces the local pH gradients and the charges in charge-carrier density on the gFET channel, which enables label-free molecular detection.

Many researchers have utilized covalent bonding with gFET channels or linker molecules to stably immobilize enzymes on the gFET [18]. For instance, the enhanced covalent bonding between graphene channels and biomolecules by weak O_2_ plasma can improve the sensing performance of gFETs [19]. However, the direct immobilization of biomolecules into the channel can potentially damage the sp^2^ structure of graphene and affect the enzyme’s bioactive site, which will cause enzyme denaturation and deter the sensing performance. For this reason, various approaches have been taken to develop an interface architecture, such as chemoresponsive polyelectrolyte thin films, to maintain the sensing performance of gFETs during enzyme immobilization. Piccinini et al. reported the layer-by-layer (LbL) assembly with functional polymer thin films on reduced graphene oxide (rGO)-FET biosensors, which enables an amplified pH response and sensitive urea detection [15]. Despite improving sensor performance, the LbL technique limits the scalable fabrication of gFETs because the technique requires a counter-polyelectrolyte and a time-consuming process for polymer deposition. In addition, Fenoy et al. reported the electropolymerization of poly (3-amino-benzylamine-co-aniline) (PABA) on rGO-FET, which provides the appropriate interfacial environment for enzyme immobilization and improves pH sensitivity [20]. Although the electrodeposition method can precisely control the thickness of polymer thin films, it still remains a challenge that a specific voltage must be applied to each device, thereby making mass-production impossible due to the large-area functionalization difficulty.

To address these issues, herein, we developed a wafer-scale fabrication of polyaniline (PAni)-grafted graphene-based FET (PGFET) for monitoring pH variation and local Ach concentration with a nanomolar limit of detection (LOD) in a flow configuration. We successfully demonstrated the scalable deposition of PAni films on the GFET using atomic force microscopy (AFM), scanning electron microscopy (SEM), high-resolution transmission electron microscopy (HRTEM), and X-ray photoelectron spectroscopy (XPS). To validate the enhanced sensing performance of the PGFET, we investigated the sensitivity and reproducibility of the PGFET compared to those of the bare-gFET, depending on pH variation. In addition, to develop an Ach biosensor, we functionalized acetylcholinesterase (AchE) on PGFET, which possesses a thin PAni film (~20 nm). The PAni film does not only provide a suitable environment for electrostatic enzyme immobilization, but it also exhibits 2.03 times improved sensitivity for Ach detection compared to the bare-gFET (~0.66%/Ach dec). Furthermore, we investigated the effects of rivastigmine, a Food and Drug Administration (FDA)-approved medication for AchE inhibition, to elucidate enzymatic drug-screening applications as a biosensor platform.

## 2. Materials and Methods

### 2.1. Chemicals and Reagents

Phosphate buffer (pH 7.4), aniline (>99%) (Ani), acetylcholine chloride (>99%), acetylcholinesterase from Electrophorus electricus (electric eel) (Type V-S, lyophilized powder, >1000 units/mg) (AchE), ethanol, hydrochloride (HCl), ammonium persulfate (APS), dopamine hydrochloride (DA), serotonin hydrochloride (SA), D-glucose (>99.5%), L-ascorbic acid (>99%) (AA), hydroiodic acid (HI), 3-Aminopropyl(diethoxy)methylsilane (APDMES), and rivastigmine tartrate (≥98%) were purchased from Sigma Aldrich. The standard liquid pH solutions were purchased from Daejung Chemicals & Metals Co., Ltd. (Siheung-Si, Korea).

### 2.2. Fabrication of PGFET and Bare-gFET

In this study, photolithography was used to fabricate both PGFET and bare-GFET devices. In detail, the bi-layer lift-off technique (2 μm lift-off resist (LOR) 5B interior layer and 3 μm photoresist AZ 601 exterior layer) was used. To serve as a source for the devices, Ti/Au (5 nm/45 nm) patterns were first deposited on thermally oxidated SiO_2_ substrates with an e-beam evaporator to serve as a source, formed via a lift-off technique. These Au-patterned SiO_2_ substrates were cleaned with a piranha solution (H_2_SO_4_:H_2_O_2_ = 3:1) and treated with oxygen plasma (100 W, 3 min) to fully activate the SiO_2_ substrate. Then, the substrates were soaked in 1% APDMES dissolved in ethanol to induce the formation of the amine group (–NH_2_) on the surface for 3 h; the functional amine group improves the adhesion between the SiO_2_ substrate and GO flakes. Subsequently, 8 mL of the GO solution was spin-coated (at 3000 rpm, for 60 s) on the APDMES-treated SiO_2_ substrate to form GO thin films. The deposited GO thin films were chemically reduced to rGO by hydroiodic acid (HI) vapor at 80 °C for 3 h. To synthesize grafted-PAni on rGO thin films, the mixture solution of APS and aniline was prepared. Specifically, 49.8 mg of APS was dissolved in 50 mL of 0.2 M HCl, followed by 20 μL of aniline addition with intensive vortexing. Immediately after the preparation of the mixture solution, the SiO_2_ wafer was loaded into a glass beaker, and then the mixture solution was directly transferred into a wafer-loaded beaker. The beaker was gently agitated (50 rpm, 25 °C) in a shaking incubator (Daihan Scientific, Korea) for 1 h, to induce the chemical polymerization of aniline from the surface of the rGO. Subsequently, the PAni-grafted rGO substrate was washed with a flow of distilled water to prevent the adsorption of the PAni precipitate on the surface. Next, the PAni-grafted rGO film was patterned with dimensions of 80 μm × 160 μm (width × length) using reactive-ion etching (RIE). The source-drain electrodes were passivated using a photoresist (SU-8 2002, Kayaku Advanced Materials Inc., Westborough, MA, USA) with a partially exposed active channel to prevent current leakages through the electrolytes. To form an aqueous environment, a microfluidic channel embedded jig was fabricated. Finally, an Ag/AgCl reference electrode (LF-2, Harvard Apparatus, Holliston, MA, USA) was utilized to directly contact the injected solution for applying a gate voltage. Note that, In the gFET fabrication process, all procedures are the same except for the polyaniline synthesis process in the PGFET fabrication.

### 2.3. Electrical Measurement and Device Characterization

The electrical response of fabricated devices was measured continuously using a semiconductor parameter analyzer (4200A-SCS, KEITHLEY, Cleveland, OH, USA) and a customized jig. The jig was composed of a microchannel-embedded polycarbonate top jig, a leakage-free Ag/AgCl reference electrode holder, and a device-loading bottom jig with a heat feedback system. For real-time measurements, the constant source-drain voltage and the electrolyte-gate voltage (V_g_) were biased at 100 mV and −300 mV, respectively. The electrical measurement was performed in a customized Faraday cage (FRD, Suwon, Korea) with a constant flow rate by a syringe pump (Fusion 200-X, Chemyx, Stanford, TX, USA).

Topographic analysis was performed via AFM (NX-10, Park systems, Suwon-si, Korea) in tapping mode to profile the thickness and roughness of our devices. For the investigation of the thickness and the conformation of the PAni-grafted rGO surface, HRTEM (JEM-ARM200F, JEOL, Peabodym, MA, USA) and FE (field emission)-SEM (JSM-6701F, JEOL, Peabodym, MA, USA) imaging were performed. The XPS spectra were investigated using K-alpha (Thermofisher, Altrincham, UK).

### 2.4. AchE Electrostatic Immobilization and the Inhibition Test of the Enzyme

A 1 mg/mL solution of AchE in a PB buffer (10 mM, pH 7.4) was prepared for the electrostatic immobilization of AchE. The PGFETs were incubated in the AchE solution for 1 h, followed by buffer washing. The enzyme-immobilized biosensors were stored at 4 °C. To evaluate the inhibition reactivity of AchE, each concentration (20 and 200 μM) of rivastigmine was dissolved in the PB buffer (10 mM, pH 7.4). The enzyme-immobilized devices were incubated in the rivastigmine solution for 30 min, and various concentrations of Ach were injected into the AchE-immobilized PGFETs, and the electrical response was measured.

## 3. Results and Discussion

### 3.1. Fabrication of PAni-Grafted gFETs (PGFETs) 

The PAni-grafted rGO was synthesized by in situ chemical polymerizations after nucleation of aniline monomer using rGO thin film as a framework. In the early stages of the polymerization reaction, the dispersed aniline monomers could be efficiently adsorbed onto the rGO substrate due to strong π-π interactions between them [21,22]. In addition, the negatively charged rGO surface electrostatically interacted with the amine group of the aniline monomer, significantly accelerating the formation of a weak charge-transfer complex [23]. Therefore, the rGO thin film provides numerous active sites for the nucleation of aniline monomers without the undesired aggregation of PAni nanorods [24]. Aniline polymerization can be confirmed by the color change from the bare rGO (dark blue) to the PAni-grafted rGO (green) in Appendix A. The uniform coating of PAni on the rGO surface can be observed in both photographic and optical microscopy images, which implicates a complete coverage of the substrate.

Based on the PAni-grafted rGO films, we successfully fabricated wafer-scale (6 in.) PGFETs and utilized them for enzyme-immobilized biosensors (Figure 1a,b). A single PGFET device photograph contained six individual PAni-grafted rGO arrays with well-passivated source-drain electrodes by a biocompatible photoresist (SU-8), presented in Figure 1c. The magnified image of the PGFET indicated that each channel was partially exposed to an analyte solution with a 20 × 160 μm^2^ area (Figure 1d). We analyzed the micro- and nano-structure of PAni-grafted rGO films by SEM and HRTEM. The SEM image of the PAni-grafted rGO showed bumpy and rough surfaces, indicating the three-dimensional granular structure of the grafted PAni with a 10 nm diameter. The cross-sectional HRTEM image of the PAni-grafted rGO film reveals that the thicknesses of rGO (black surface) and grafted PAni layers (white surface) are approximately 10 nm and 20 nm, respectively. The results clearly indicate that the hierarchical polymerization of PAni had been accomplished on the rGO surface.

To test the sensing performances of PGFET biosensors, we immobilized the AchE onto the biosensor’s surface, and the enzymatic reaction was confirmed by measuring the changes in its electrical characteristics (Figure 1b). In detail, the AchE catalyzes the hydrolysis of Ach to form acetic acid and chlorine, which induce the generation of protons and the local pH variations on the surface of the PGFET. The Ach hydrolysis can be transduced into the changes in the relative conductivity of the PGFET due to the charge transfer between the rGO channel and PAni nanorods. By quantifying the enzymatic reaction, we developed a real-time Ach-sensing platform in a flow configuration.

### 3.2. Characterization of PGFETs

To investigate morphological characteristics of both the bare-gFET and the PGFET, high-resolution topography images were acquired by AFM. The AFM image of the gFET showed a relatively smooth structure with typical stacks and grain boundaries of rGO flakes (Figure 2a and Appendix A). Conversely, the image of the PGFET showed an uneven granular structure due to the hierarchically grown PAni on the rGO surface (~20 nm), which is consistent with the SEM and HRTEM images (Figure 1e,f). We further analyzed surface roughness and height distributions of both FETs (bare-gFET and PGFET). The root-mean-square (RMS) surface roughness of the PGFET (6.104 nm) was significantly increased compared to that of the bare-gFET (0.729 nm) (Appendix A). From the height distributions of the bare-gFET and PGFET, we assessed the peak shift of height distribution after PAni polymerization from 0.53 ± 0.99 to 20.65 ± 5.67, which is consistent with the HRTEM image. Collectively, the results demonstrated the growth of PAni on the overall rGO surface, as illustrated in Figure 2c. 

X-ray photoelectron spectroscopy (XPS) spectra also confirmed that the grafted-PAni entirely covered the rGO surface (Figure 2d). In the PAni-grafted rGO spectrum, N 1s peaks appeared at 400 eV, and the intensity of O 1s peaks decreased, compared to that of the bare-gFET, indicating that the grafted PAni concealed the functional group of the rGO surface. Figure 2e shows the N 1s peaks of the PGFET and bare-gFET. The N 1s peak of the PAni-grafted rGO was deconvoluted into a positively charged nitrogen (N^+^) atom with a binding energy of 399.52 eV, imine nitrogen (–N=) with a binding energy of 397.20 eV, and amine nitrogen (–NH–) with a binding energy of 398.38 eV. Conversely, the spectrum of bare-gFET shows no significant peaks in N 1s. 

The transfer characteristics (drain current (I_ds_) vs. gate voltage (V_g_)) of the bare-gFET (gray curve) and PGFET (red curve) with a constant drain bias of 100 mV are illustrated in Figure 2f. The bare-gFET and PGFET exhibited typical ambipolar transfer characteristics and a Dirac point (V_Dirac_), which is a specific gate voltage at which the source-drain current reaches the minimal value. In a transfer curve, the V_Dirac_ of the PGFET shifted negatively, by 73 mV, compared to that of the bare-gFET, due to an electrostatic gate effect of a positively charged polymer on the rGO channel. However, the slope of the transfer curve, referred to as the transconductance, remained relatively constant compared to that of the bare-gFET, demonstrating that the solution-based polymerization of PAni does not impair the performance of the gFET. Moreover, we confirmed that the leakage current is negligible (only a few nA) compared to the source-drain current (about dozens of μA), indicating that the PGFETs were completely passive, as illustrated in Appendix A.

### 3.3. pH-Sensing Properties of PGFETs

Before validating the sensing performances of the PGFET, we constructed a customized jig system including microfluidic channels and a heat feedback system. (Figure 3a). In detail, the top jig contained an Ag/AgCl gate electrode and microfluidic channels for real-time monitoring of electrolyte-gated FETs (EGFETs). Furthermore, we constructed a thermostatic condition (37 °C) utilizing a temperature feedback system on the bottom jig because pH- and enzymatic reactions crucially rely on temperature [25,26]. We applied a constant flow rate (300 μL/min) into the microfluidic channels and monitored the real-time pH responses of the bare-gFET and PGFET under a modest gate potential (V_g_ = −0.3 V) to prevent undesired chemical reactions [20]. The results show that the relative changes in the source-drain current values (ΔI_ds_/I_0_) gradually decreased in both FETs (bare-gFET and PGFET) depending on the pH variation from 9 to 3 (Figure 3b). This is because the applied gate voltage was below the Dirac point, which is in the hole regime. From the results of Figure 3b, we analyzed ΔI_ds_/I_0_ as a function of pH values from 9 to 4 using linear regression curves (Figure 3c). The extracted slopes of each curve were 2.68 and 1.81%/pH, which indicated that the sensitivity of the PGFET significantly improved (~48%), compared to that of the bare-gFET. The improved sensitivity of the PGFET arises from the protonation and deprotonation of PAni-grafted rGOs depending on the surrounding pH condition [27,28]. Specifically, the grafted PAni can be converted from a dielectric emeraldine base (EB) to a conductive emeraldine salt (ES) with the pH decrement. The protonation and deprotonation of the PAni can be transduced into the changes in the drain current of the PGFET. 

Next, we conducted the repeatability test by measuring the real-time responses of the PGFET and bare-gFET with the pH cycling at various pH values (i.e., 3, 5, 7, and 9), as illustrated in Figure 3d and Appendix A. The results showed that the PGFET maintains a signal consistency in ΔI_ds_/I_0,_ depending on three pH cycles, which demonstrated that the PGFET exhibits excellent reproducibility and repeatability. We further analyzed pH-hysteresis, which describes the signal difference between the initial and the final electrical responses of the bare-gFET and the PGFET for each pH cycle, as illustrated in Figure 3e. The results revealed that the hysteresis in ΔI_ds_/I_0_ of the PGFET (5.57 × 10^−7^%) was approximately seven orders of magnitude lower than that of the bare-gFET (2.40%), indicating full recovery to the baseline without any hysteresis. These results suggest that the PAni-grafted rGO acts as an interface providing a high pH sensitivity and repeatability. Altogether, our PGFET has great potential as a biosensing platform for monitoring local pH changes induced by enzyme-catalyzed hydrolysis.

### 3.4. Quantification of Acetylcholine (Ach)

Beyond studying the pH-sensing performance of the PGFET, we expanded its application to quantify Ach by the enzyme-immobilization on the PAni-grafted rGO. The enzyme (AchE) was functionalized on the PGFET via electrostatic interaction between the PAni-grafted rGO and AchE without enzyme denaturation or malfunction by disrupting the functional groups [29]. To validate the immobilization of the AchE, we performed the topographic analysis of the AchE using AFM (Appendix A). From the results, the AchE has about 8.90 nm height on the overall PGFET surface.

Next, we assessed the real-time responses in the ΔI_ds_/I_0_ curve with an increasing Ach concentration and both AchE-immobilized biosensors (PGFET and gFET) illustrated in Figure 4a. The ΔI_ds_/I_0_ curves of both AchE-immobilized biosensors decreased depending on various Ach concentrations from 108 nM to 2 mM. This is because the AchE decreased the local pH by the enzymatic hydrolysis of Ach, transducing into decreased I_ds_ under the gate voltage biased in the hole regime. In contrast, the PGFET without any AchE immobilization shows no signal differences depending on various Ach concentrations, which indicates that the signal responses only occurred by the enzymatic reaction of AchE. To analyze the sensitivity of the AchE-immobilized biosensors, we plotted linear curves between ΔI_ds_/I_0_ in both AchE-immobilized biosensors and the logarithmic Ach concentrations, as illustrated in Figure 4b. The sensitivity of the PGFET was obtained as 1.34%/Ach dec from the slope of the PGFET, which was significantly improved compared to that of the bare-gFET (0.66%/Ach dec). This is because the hierarchical structure of grafted PAni provides a large surface area compared to that of a bare rGO surface, which facilitates the electrostatic bindings for AchE immobilization. The limit of detection (LOD) of the PGFET was calculated as 3.3 σ/S where σ is the standard deviation from the blank test (buffer injection), and S is the slope of the calibration curve, yielding a value of 72.3 nM. Figure 4c illustrates the I_ds_ responses of each PGFET depending on various Ach concentrations, which supported a good reproducibility of PGFETs regardless of the initial resistance of each PGFET. The key analytical parameters of the present biosensor are compared to those of other recently described enzymatic FETs for acetylcholine detection in Appendix A. Our PGFETs have outstanding characteristics, including an extensive dynamic range, excellent sensitivity, and a low LOD.

To investigate the enzymatic kinetics of an AchE immobilized on the PGFET, we fitted the normalized ΔI_ds_/I_0_ of the PGFET to the Michaelis–Menten equation (Appendix A). We extracted the Michaelis–Menten constant (K_M_) of 20.80 μM from the equation, which means the AchE on the PGFET possesses 2–10 times higher activity than that from previous research [20,30]. We presumed two reasons for the higher activity of the AchE in the PGFET: (1) The AchE is electrostatically bound to the grafted PAni without denaturation or malfunction. (2) The customized jig provides a well-confined thermostatic condition for optimal enzymatic reactions at approximately 35 °C [31,32]. 

The selectivity of the PGFET was assessed using various interfering molecules, including glucose, ascorbic acid (AA), dopamine (DA), and serotonin (SA), at physiological concentrations [33,34]. We sequentially injected each interfering molecule into the AchE-immobilized PGFET with a constant flow rate (300 μL/min) and compared the signal response (ΔI_ds_/I_0_) with 16 μM Ach. The PGFETs barely reacted with the interfering molecules (up to −0.26%), whereas the significant responses of PGFETs were observed with 16 μM Ach (~−2.85%). Altogether, the results strongly supported the excellent selectivity of our PGFETs.

### 3.5. Inhibition Test of Acetylcholinesterase (AchE)

AchE inhibitors are used as a standard treatment for neurotransmitter imbalance in the brain by reducing an excessive Ach hydrolysis. Rivastigmine is a phenyl-carbamate derivative, which is the only FDA-approved inhibitor for lowering both the activities of AchE and butyrylcholinesterase in the brain. While Ach is dissociated almost immediately after the hydrolysis, rivastigmine is classified as a pseudo-irreversible inhibitor by binding to the anionic and esteratic sites of AchE up to 10 h [35]. We utilized rivastigmine to further investigate the feasibility of our PGFET biosensor for drug-screening applications. We firstly pre-incubated AchE-immobilized PGFETs in 0, 20, and 200 μM of rivastigmine for 30 min to fully induce non-competitive inhibition of AchE [36]. After the incubation, we injected various Ach concentrations into AchE-immobilized PGFETs with a constant flow rate (300 μL/min) and monitored the real-time responses (ΔI_ds_/I_0_) of each PGFET (Figure 5a). At low Ach concentrations (<80 μM), PGFETs with rivastigmine showed negligible responses. In contrast, at a higher Ach concentration (400 μM), the signal responses (ΔI_ds_/I_0_) of PGFETs with 20 μM and 200 μM of rivastigmine appeared. For evaluating the inhibition effects of rivastigmine, we compared ΔI_ds_/I_0_ of PGFETs with various conditions (0, 20, and 200 μM of rivastigmine) depending on various Ach concentrations (Figure 5b). From the results, the inhibition rates of 20 μM and 200 μM rivastigmine were calculated as 69.8% and 86.8%, respectively, at a concentration of 400 μM Ach (Figure 5c). Considering that AchE inhibitors are a standard treatment for neurodegenerative diseases, the PGFET biosensors have the potential as a drug-screening platform for monitoring the efficacies of AchE inhibitors. 

## 4. Conclusions

We demonstrated the scalable fabrication of polyelectrolyte-grafted gFETs based on the chemical in situ polymerization of PAni on the rGO surface and its versatility for biosensing applications. Compared to the bare-gFET, the PGFET significantly improved pH sensitivity (~48%) without any pH-hysteresis due to the pH-dependent transition from ES to EB forms of grafted PAni. Based on this mechanism, the PGFET allowed for the real-time monitoring of the Ach hydrolysis ranging from 108 nM to 2 mM with significantly improved sensitivity (~103%) compared to the bare-gFET. This is because the hierarchical structure of the grafted PAni enhanced both local pH sensitivity and electrostatic AchE immobilization on the PGFET. Moreover, the PGFET exhibited a great selectivity against various interferences known to exist in the extracellular milieu with rapid response time and good reliability. Finally, considering the monitoring inhibitory effects of the FDA-approved AchE inhibitor (rivastigmine), our PGFET has the potential to be applied as a drug-screening platform for monitoring the efficacies of enzyme inhibitors.

## Figures and Tables

**Figure 1 biosensors-12-00279-f001:**
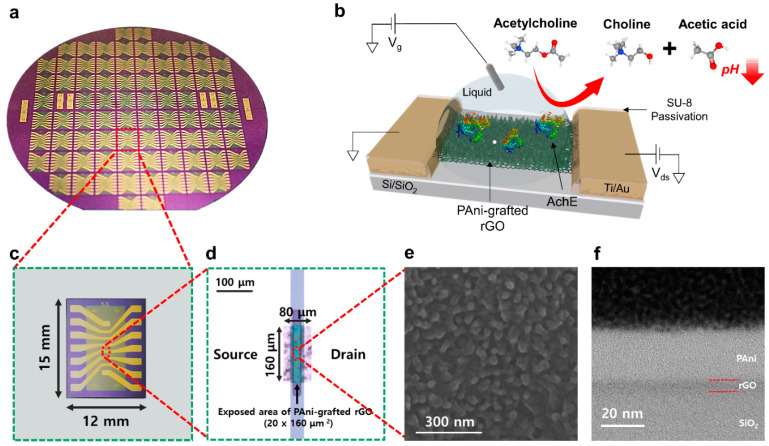
Wafer-scale fabrication of PGFETs for biosensing applications. (**a**) Photograph image of the fabricated PGFETs on a 6-inch wafer. (**b**) Schematic illustration of a PGFET biosensor for monitoring the enzymatic reaction of AchE. (**c**) Photographic image of a single PGFET device. (**d**) Magnified optical microscopy image showing the partially exposed area of PAni-grafted rGO. (**e**) SEM image showing a part of the PGFET channel with the granular structure of grafted PAni. (**f**) TEM image presenting the cross-sectional profile of fabricated PAni-grafted rGO.

**Figure 2 biosensors-12-00279-f002:**
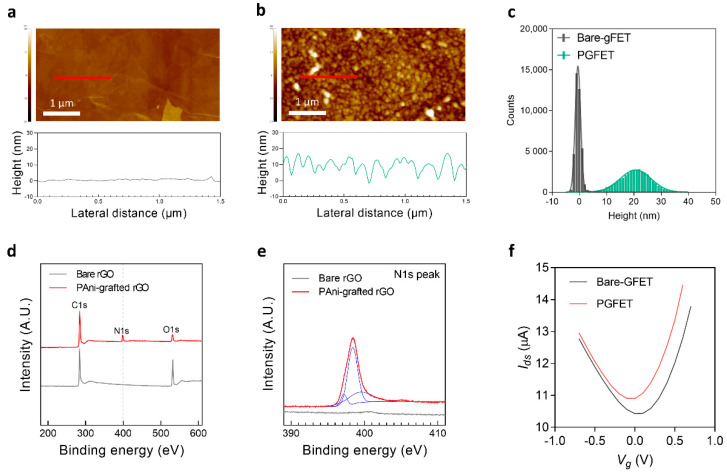
Surface characterization of the bare-gFET and PGFET. The AFM images of the bare-gFET (**a**) and PGFET (**b**) and their corresponding height profiles. (**c**) The height distributions of the bare-gFET and PGFET in a 1 × 1 μm^2^ area. (**d**) XPS analysis of the bare rGO and PAni-grafted rGO. (**e**) High-resolution N 1s XPS spectrum of the bare rGO and PAni-grafted rGO. (**f**) I_ds_ vs. V_g_ curves for a bare-gFET and PGFET.

**Figure 3 biosensors-12-00279-f003:**
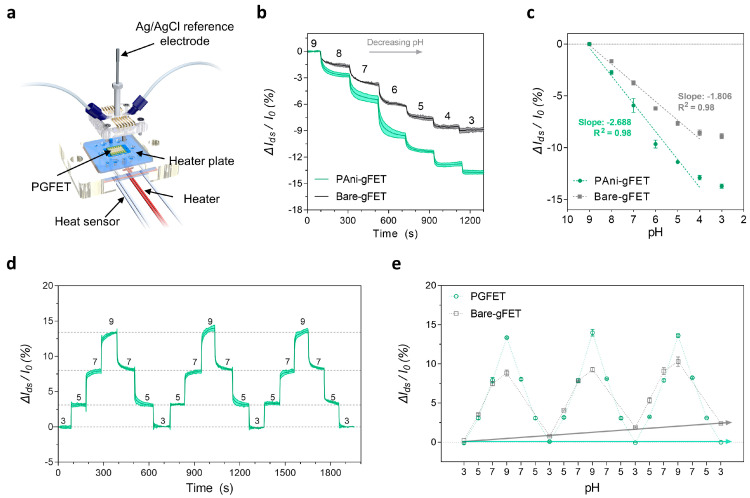
pH-sensing performance of the bare-gFET and PGFET. (**a**) Schematic illustration of the customized jig for real-time monitoring in a flow configuration. (**b**) The real-time monitoring depending on the pH variation from 9 to 3 in the bare-gFETs and PGFETs. (**c**) The linear plots of the bare-gFET and PGFET for the analysis of pH sensitivity. (**d**) The real-time output responses of PGFETs for three pH cycles. (**e**) The output responses of bare-gFETs and PGFETs as a function of cyclic pH variations for representing the pH-hysteresis of each device. The light band around the curves represents pointwise 95% confidence intervals derived from *n* = 3 independent measurements.

**Figure 4 biosensors-12-00279-f004:**
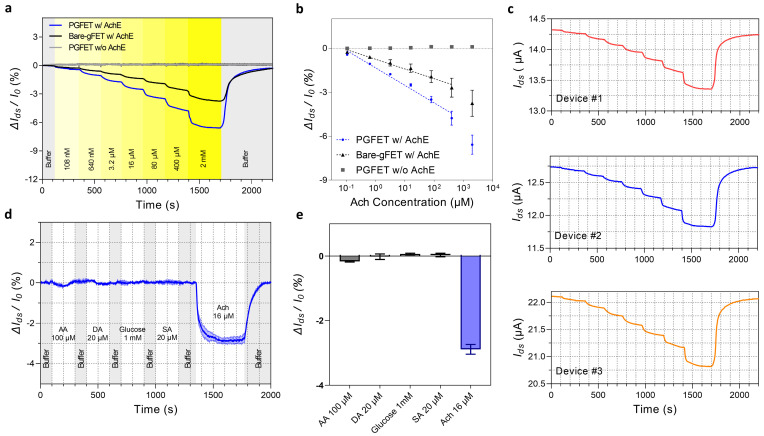
Ach-sensing performance of an AchE-immobilized bare-gFET and PGFET. (**a**) The real-time responses with increasing Ach concentrations in AchE-immobilized PGFETs (blue line), AchE-immobilized bare-gFETs (black line), and PGFETs without AchE immobilization (gray). (**b**) The plots fitted by linear regression as a function of Ach concentrations from 108 nM to 400 μM for the sensitivity analysis of each biosensor. (**c**) The real-time I_ds_ responses of individual PGFETs depending on various Ach concentrations. (**d**) The real-time responses of the AchE-immobilized PGFET with various interfering molecules and Ach (16 μM). The light band around the curve represents pointwise 95 % confidence intervals derived from *n* = 3 independent measurements. (**e**) Bar graphs representing the comparison between the signal of the target analyte and interferences. The average ± standard deviation was calculated from *n* = 3 independent samples.

**Figure 5 biosensors-12-00279-f005:**
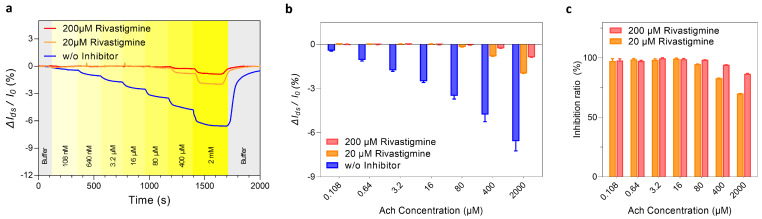
Inhibition effects of rivastigmine on AchE activity tests. (**a**) The real-time responses of AchE-immobilized PGFETs with various rivastigmine concentrations (0, 20, and 200 μM). (**b**) The ΔI_ds_/I_0_ responses for a quantitative analysis of inhibition effects depending on rivastigmine concentrations (0, 20, and 200 μM). (**c**) Inhibition rates for different concentrations of rivastigmine. The average ± standard deviation was calculated from *n* = 3 independent samples.

## Data Availability

Not applicable.

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
