# Peer review of "Scalable Functionalization of Polyaniline-Grafted rGO Field-Effect Transistors for a Highly Sensitive Enzymatic Acetylcholine Biosensor"

_biosensors, 2022, doi:10.3390/bios12050279_

Round 1
Reviewer 1 Report
The authors propose a wafer-scale fabrication of polyaniline (PAni)-grafted graphene based field-effect transistors (PGFET) and their biosensing applications for sensitive real-time monitoring of Ach in flow configuration. The paper is well written and the results are sound. I would like to suggest it published on Biosensors after Minor revision.
- The electrical performance of the FETs without normalization should be given in fig. 2f.
- Although the fabrication process is wafer scale, the author should show the statistical electrical performance after the fabrication and modification process to show the advantages and uniformity of this wafer scale fabrication process.
- The description of “Fabrication of PGFET” seems to be presented twice in the manuscript. The author may refine and simplify the description in part 2.2 and part 3.1.
Reviewer 2 Report
Park et al. developed polyaniline (PAni)-grafted graphene-based field-effect transistors (PGFET) for the enzymatic detection of acetylcholine. Various characterization techniques are used to confirm the deposition of PAni on GFET. The work is well described all necessary supporting results for the findings. The reviewer suggests the acceptance of the manuscript. However, there are a few comments that need to be addressed.
- Line no. 112, 164; correct the SiO2 to SiO2. There are a few typo errors in the manuscript. Please pay attention to those.
- The X and Y-axis labelling font sizes are not uniform in most of the figures. The reviewer suggests uniformity in all the figures.
- The sensing platform is composed of the thermostat, what will be the effect of varying temperature on detection? Please describe.
Reviewer 3 Report
The manuscript titled ‘Scalable functionalization of polyaniline-grafted rGO field-effect transistors for a highly sensitive enzymatic acetylcholine 3 biosensor’ is used for tracing acetylcholine. A polyaniline-grafted rGO field-effect transistors has been fabricated to detect ach and demonstrated PGFET has great potential as a real-time drug-screening platform.
The work is good and interesting. The sensitivity of the sensor is appreciable and have been tested for drug screening as well.
However, there are similar reports on detection of acetylcholine using PANI gratfed rGO platforms. What is the major innovation in this work as compared to other reported platforms?
What is the assay time for sensing ach and drug screening?
PANI and rGO are complex molecules with high binding affinity towards silicon and gold. How did you manage to avoid non specific binding towards the target enzyme and achieve high sensitivity?
Can we measure in situ monitoring of ach level in patient samples?
Please add a comparison table of sensing parameters of other reported ach sensors.
